# Objectively Assessed Physical Activity of Preschool-Aged Children from Urban Areas

**DOI:** 10.3390/ijerph17041375

**Published:** 2020-02-20

**Authors:** Jarosław Herbert, Piotr Matłosz, Justyna Lenik, Agnieszka Szybisty, Joanna Baran, Karolina Przednowek, Justyna Wyszyńska

**Affiliations:** 1Institute of Physical Culture Sciences, Medical College, University of Rzeszów, 35-959 Rzeszów, Poland; pmatlosz@ur.edu.pl (P.M.); ju_lenik@ur.edu.pl (J.L.); aszybisty@ur.edu.pl (A.S.); karprzed@ur.edu.pl (K.P.); 2Institute of Health Sciences, Medical College, University of Rzeszów, 35-959 Rzeszów, Poland; jbaran@ur.edu.pl (J.B.); jwyszynska@ur.edu.pl (J.W.)

**Keywords:** accelerometery, objective monitoring, young children, physical activity, preschool

## Abstract

Little is known about physical activity (PA) of preschool-age children in Poland through the course of the day. PA monitoring using an accelerometer increases the reliability of measuring daily PA levels and offers a reasonable compromise between accuracy and feasibility of measurement. The aim of the study was to determine the level of physical activity of preschool children (aged 5–6) on the basis of moderate to vigorous physical activity (MVPA) index and the number of steps. The physical activity of preschool children was assessed using accelerometery (ActiGraph) in 371 children for up to seven days. The normality of distribution was assessed using the Shapiro–Wilk test. The Mann-Whitney U-test and Kruskal-Wallis test were used to assess the significance of differences. The study group children had an average age of 5.4 years (± 0.6). Boys and girls showed a different level of MVPA index. The results significantly improve the current knowledge of PA in Europe. Promoting active lifestyles in children should be one of the health priorities in developed countries.

## 1. Introduction

Approximately 1.02 million Polish children aged 3–6 years attend kindergartens, and this number has increased significantly over the last decade (by about 25%) [1]. On average, children spend 35 h a week in kindergartens or day-care centers [1]. As the number of children in kindergartens increases, it is likely that their level of physical activity (PA) decreases, which affects their energy balance and exposes them to the risk of becoming overweight. Preschool-age children are commonly believed to be physically active [2].

Regular PA of appropriate intensity provides many benefits to the health of children. Higher PA levels are associated with better body weight composition, bone condition, mental health and school performance [3,4,5,6]. Moreover, in early childhood PA, not only serves to develop and maintain normal healthy habits [7], but also plays an important role in the development of cognitive functioning, socialization and emotional well-being [8].

A decrease in active transport to school/kindergarten (bicycle, skateboard, rollerblades, etc.) causes PA deficits in children and adolescents [9,10], as well as an increase in the time spent in front of digital device screens [11,12].

Currently, public health guidelines [13] on PA place special emphasis on the population of children (6–11 years old) and adolescents (12–19 years old). These guidelines usually address the frequency, time and intensity of PA. Therefore, accelerometers are gaining increasing interest and popularity in studies as an objective tool for measuring daily PA.

PA monitoring using an accelerometer provides a reasonable compromise between accuracy and feasibility of measurement [14]. In addition, it increases the reliability of the measurement of the daily PA level [15].

The World Health Organization (WHO) and public health authorities around the world recommend that children aged 5–17 years take at least 60 min of Moderate to Vigorous Physical Activity (MVPA) every day in order to gain optimal health benefits [13]. Regarding the number of steps, various governmental organizations around the world follow the 12,000 steps per day recommendation (for children) as a benchmark for the minimum level of PA. This guideline for the number of steps is endorsed by the WHO, National Heart Association of Australia, US Centres for Disease Control and Prevention and American Heart Association to improve overall health [16,17]. Although there is ample evidence that PA provides significant health benefits to young people, the activity of most children and adolescents in developed countries does not meet the PA guidelines [18].

Increasing the PA levels in children should result in a reduction in sedentary time, forming the habit of being active and thereby increasing the likelihood that they will be physically active and fit during the rest of their childhood, adolescence, and adulthood [19].

Pre-schoolers should accumulate at least 60 min of structured PA each day [20]; however, O’Dwyer et al. [21] suggest that school represented an environment that promoted a sedentary lifestyle. In this regard, Barbosa et al. [22] showed that children attending preschool spend most of the day in sedentary behavior.

The related literature shows that the level of physical activity can vary significantly in childcare facilities. This difference may be due to differences in policies, traditions, programs of study, use of teachers’ time and qualifications; but it also may be due to differences in the physical environment in childcare facilities [23,24].

Preschool children are of particular concern as they are in a critical period of growth and development. Therefore, understanding the association between physical activity and body mass index (BMI) in this group, it is important to inform about future strategies for the prevention of obesity among others.

This is the first study carried out on the physical activity of preschool-age children in Poland observed throughout the day on a large group of respondents.

The aim of the study was to determine the level of physical activity of preschool children (aged 5–6 years) on the basis of the MVPA index and the number of steps. In addition, it assessed whether preschool children were sufficiently physically active, in accordance with the WHO guideline (a minimum of 60 min per day of MVPA and/or 12,000 steps per day).

Based on previous studies [18] we hypothesize that most of the subjects do not meet WHO physical activity recommendations. We also assume that gender and BMI will significantly differentiate the physical activity level of preschool children.

## 2. Materials and Methods

The study was approved by the Bioethics Committee of the University of Rzeszów (no. 2017/01/05). Before the study was initiated, written consent for participation was obtained from the children’s parents. The anthropometric measurements were carried out in kindergartens, in a separate room. All the measurements were taken between 8:00 a.m. and 10:00 a.m. With the help of tutors/teachers, all participants received comprehensive information about the study.

### 2.1. Participants

The study covered a total of 371 children aged 5–6 years, attending public kindergartens in Rzeszów (Rzeszów is a city in Southeastern Poland with a population of approximately 200,000), born in the years 2012–2013 (n = 371).

We recruited a representative group of children in their preschool year (n = 371) by recruiting from selected postal sectors in the Rzeszów area.

The study was conducted in the 2017/18 school year. An invitation to participate in the study was sent to 462 parents of children attending public kindergartens. Out of this number, 452 parents agreed to the participation of their children in the study.

Of these, 81 were excluded from the study for the following reasons: removal of the accelerometer at any time during the study period; the device showing a mechanical error and/or operator error (incorrect epoch length, incorrect anthropometry or incorrect participant identification) (n = 66); and refusal to participate (n = 15). Finally, 371 children were included in the analysis.

### 2.2. Anthropometric Measurements

Body height was measured with an accuracy of 0.1 cm using a Seca 213 portable stadiometer. The measurement was taken in a vertical position, barefoot. Bodyweight was assessed with an accuracy of 0.1 kg using a body composition analyzer (BC-360, Tanita). The body mass index (BMI) was calculated as body weight (kg)/height (m) squared. Based on the BMI values, the BMI percentiles were calculated by reference to the Polish centile grids [25]. Based on the BMI percentile values, a participant’s body mass categories were determined as follows: underweight (<5th percentile), healthy body mass (between 5th and 85th percentile), overweight (BMI ≥85th percentile and < 95th percentile), or obese (≥95th percentile) [26]. All the measurements were taken on fasting status, early in the morning, before the accelerometer was set up, according to the manufacturer’s guidelines.

Anthropometric measurements were made in kindergartens in separate rooms by experienced employees of Rzeszów University. The children were without shoes and in light clothing. The data were collected in Rzeszów University’s database.

### 2.3. Physical Activity

Currently, accelerometers are used in many studies on the level of physical activity. This study used the ActiGraph WGT3X-BT (Pensacola, USA) triaxial accelerometer [27].

The accelerometer was placed on the waist using a flexible strap, above the right hip bone, in order to measure the participant’s motion rate and frequency. After the end of the recording, the sensor was connected to a computer via a mini-USB to transfer the data. During initialization, information such as the participant’s name, sex, height, weight, and race was acquired. The participants were instructed to wear the accelerometer for seven consecutive days, 24 h a day, five days a week and two days on the weekend. The data were collected in 5-s epochs [28]. Non-wear time was defined as 60 min of consecutive zeros, allowing for 2 min of non-zero interruptions [29].

Wear time of ≥500 min/day was used as the criterion for a valid day, and ≥ 4 days were used as the criteria for a valid 7-day period of accumulated data [29]. Data from 7:00 a.m. to 11:00 p.m. were used for analysis [30]. ActiGraph data were analyzed with Actilife 6.0. (ActiGraph LLC, Pensacola, FL, USA).

The cut-off points from Evenson et al. were selected to determine the time spent on MVPA level (>2296 counts per minute—CPM). The cut-off points were: Sedentary: 0–100 CPM, Light: 101–2295 CPM, Moderate: 2296–4011 CPM, Vigorous: 4012–∞ CPM [31].

The participants complying with the minimum 60 min of MVPA per day requirement met the guidelines, while the participants who did not meet this number (<60 min) were regarded as inactive [32].

Daily step count was calculated as the mean daily step count from all valid days.

Accelerometers were placed on the children in the morning in kindergartens. The children were informed about how the accelerometer works.

### 2.4. Statistical Methods

The normality of the distribution was assessed using the Shapiro-Wilk test. None of the analyzed variables showed compliance with a normal distribution and, therefore, non-parametric tests were applied. The data were presented as the mean ± standard deviation for continuous variables, as well as the number of participants and percentage fractions for nominal variables. Generalized linear models were used to investigate factors that predicted levels and change in MVPA and steps counts. The study assumed a significance level of α=0.05. The statistical analysis was conducted using the Statistica 13.0 package (Dell 2016) [33].

## 3. Results

Table 1 presents the descriptive characteristics of the study population. The study group consisted of 371 participants (185 girls and 186 boys). The participants had an average age of 5.4 years (± 0.6). Only 96 (25.9%) of the participants followed the recommendation of at least 60 min of MVPA per day and only 21 (5.7%) met the requirements of at least 12,000 steps daily.

Table 2 presents a detailed analysis of PA parameters by sex. It also shows the MVPA index level (on average, 48 ± 21 min) and the number of steps per day (8800 ± 2066 per day). Statistically significant differences between the boys and girls were found in three out of six PA indices.

The analysis showed that the boys had higher total PA levels and spent more time in moderate activities (a mean difference of 59 min; <0.0001) and MVPA (a mean difference of 9 min; *p* = 0.0001), and took more steps per day (a mean difference of 393; *p* = 0.0453).

Table 3 presents the MVPA classifications by sex, BMI classification, and Step counts. Boys and girls showed a different level of MVPA index (*p* = 0.0001; odds ratio of 0.37). The number of boys who met the recommendations of MVPA classifications was more than twice higher with regards to girls. The division into BMI classification clearly shows that the total number of 296 respondents are in the healthy body mass group, and the statistical difference is *p* = 0.0045.

The analysis of step counts classification shows the appropriate amount of MVPA and step counts has been met by a small group of 5.1% (*p* = 0.0001; odds ratio of 33.68).

Analyzing the data from Table 4 it was found that only the gender and BMI classification associated with MVPA in the full multilevel analysis were identified as determinants on the final model. MVPA was positively associated with age, body height and BMI classification. An identical phenomenon was observed regarding step counts. A negative association was noted for gender and body mass (MVPA and step counts). Statistical significance was noted only for gender and BMI classification in MVPA.

## 4. Discussion

The major strength of this study is the provision of objective physical activity data from children. However, almost 74.1% of the participants did not meet the international guidelines for a minimum MVPA level. This is one of the first large-scale studies in Poland, where a direct observation system was used to describe the PA level of children attending kindergarten.

Physical activity differed slightly between each sex. This may be due to sex-dependent motor skills [34].

PA is an important factor in decreasing cardiometabolic risk factors in children, but most children are not meeting the recommendations for health-enhancing daily PA [35,36].

A total of 25.9% of the participants met the recommended MVPA level and showed a different MVPA index level between the sexes (boys 17.5%; girls 8.4%).

In our research, the children spent an average of 4053 min/week in sedentary activities. This gives an average of 579 min/day, or 10 h/day.

Based on the analysis of our own study results, we found that the boys were more active than the girls (boys 52 ± 23 min and girls 43 ± 19 min of MVPA–*p* = 0.0001). Similar results were obtained by O’Dwyer et al. [21] and Jago et al. [37]. In a study of five-year-olds from Qatar, the average MVPA level was 26.0 ± 8.6 and 23.1 ± 8.8 min per day for boys and girls, respectively [38].

Our findings, therefore, align with literature and support the idea that preschool-aged boys are more active than their female counterparts [39,40].

A much higher result was presented by Leeger-Aschmann et al. [41], indicating that the average level of MVPA index was 86 min for a study population of 394 children from Switzerland aged 3.9 ± 0.7 years.

One of the largest scientific reports analyzing the MVPA index level in preschool children is a systematic review by Ravagnani et al. [42]. The authors analyzed 33 scientific reports, and the MVPA index was analyzed for a total of 12,178 preschool children. The mean MVPA value for this population ranged from 21.1 (Puyau’s cut-off point) to 288.6 (Freedson’s cut-off points) min/day. In all the studies, the weighted average of MVPA min/day for boys and girls was 102.7 min. In short, the MVPA index of preschool children in the study did not reach a very impressive result.

The analysis by Vincent and Pangrazi [43] was one of the first studies to evaluate a large group of pupils aged 6–12 years (n = 711). The authors suggested a minimum of 11,000 steps for girls and 12,000 steps for boys per day as a basis for physical activity. On the other hand, Tudor-Locke et al. [44] proposed for children aged 6–11 years a minimum of 12,000 steps for girls and 15,000 steps for boys, and for children aged 4–6 years: 10,000–14,000 steps/day and for adolescents aged 12–19 years: 10,000–11,700 steps/day [45].

Despite the fact that only 5.9% of the participants followed the recommendations of 12,000 steps a day, the results of our own studies do not differ significantly from those of other authors. Taking into account the number of steps (boys 8996 ± 2322; girls 8603 ± 1758), our participants are at an average level, compared to the other results. For example, in a study of five-year-olds from Greece, the number of steps were as follows: boys 10,399 ± 2320 and girls 9598 ± 1788, [46] while studies in Australia, show the number of steps of pre-schoolers were: boys 12,014 and girls 9762, [47] and the number of steps of preschool children from Belgium were: boys 10,983 ± 1720 and girls 8210 ± 2170 [48].

Adams et al. [49] found that in most cases the standard of 12,000 steps per day was too high. Tudor-Locke et al. [45] in their topic-related literature review suggested that adolescents should achieve at least 10,000 steps/day.

It is important to set an attainable step count target, although it should be high enough so that not all preschoolers are categorized as sufficiently active because literature clearly shows an increase in overweight and obesity prevalence in preschool children [50]. Studies with direct observation show low levels of PA in pre-schoolers [51,52,53] and it is logical to assume that additional health benefits can come from accumulating more steps per day.

It is worth noting that almost all of the subjects who met the recommendation of 12,000 steps (5.9%) also met the MVPA 60 min. minimum MVPA (5.1%) (Table 3).

The use of accelerometers in research is popular, and studies on different levels of physical activity measured as the number of steps help to set goals that can affect children’s PA levels and aerobic performance.

This study featured a number of strengths and limitations, which are worthy of note. The main strength of the study is the large sample size of 371 pre-schoolers with valid accelerometer data for a minimum of four complete days (two weekdays and two weekend days), and reliable direct observation made possible by the use of the accelerometer. It is the first objective study on the PA level on pre-school children in Poland in such a large group. The results may be limited to some extent, as all the kindergartens were located in a single metropolitan area, and the study sample was not an ideal representation of the population of children attending kindergarten. It did not include children from rural areas and other cities. However, it is worth emphasizing that there were no interventions in children’s PA during the study.

In conclusion, promoting active lifestyles in children should be one of the health priorities in developed countries. In the future, a similar study should be carried out on a population that includes children from rural areas and other cities.

## 5. Conclusions

The results significantly improve the current knowledge of PA in Poland. In our study, boys were more active than girls. The hypothesis was confirmed because the research showed that BMI and gender significantly differentiate the level of physical activity of preschool children.

The minority of the study group (25.9%) followed the recommendation of at least 60 min of MVPA per day, while 94.3% did not achieve the recommended 12,000 steps per day. Increasing numbers of detailed reports on the patterns that occur each day of the week can help public health strategists, teachers and, above all, children’s caregivers to develop more detailed interventions in order to increase PA levels in children throughout the day. The study involved children in urban areas, not in rural areas. The authors suggest that children from kindergartens should join programs that increase physical activity. Such programs have been introduced successively in the city of Rzeszów for several years. It may be worth encouraging parents to provide opportunities for physical free play as promoted by Gray [54] and Brown [55]. Parents’ involvement is observed as essential [56,57].

Health education for parents is needed, both in order to learn how important adequate PA is for a child’s health and development, as well as to learn strategies for stimulating children’s PA [58].

## Figures and Tables

**Table 1 ijerph-17-01375-t001:** Characteristics of children.

Variable	Values
Gender ^a^	
Girls	185 (49.9)
Boys	186 (50.1)
Age ^b^ (years)	5.4 ± 0.6
Body mass ^b^ (kg)	21.3 ± 3.8
Body height ^b^ (cm)	116.2 ± 6.2
BMI percentile ^b^	46.5 ± 31.0
Body mass classification ^a^	
Underweight	30 (8.1)
Healthy body mass	296 (79.8)
Overweight	28 (7.5)
Obese	17 (4.6)
MVPA [minutes/day] classification ^a^	
≥60	96 (25.9)
<60	275 (74.1)
Step Counts classification ^a^	
≥12000	21 (5.7)
<12000	350 (94.3)

Data are expressed as ^a^—n (%); ^b^—x̅ (SD).

**Table 2 ijerph-17-01375-t002:** Physical activity by sex.

	Total (n = 371)	Boys (n = 186)	Girls (n = 185)	Δ_M-F_	*p*
Sedentary (min)	4053 ± 1431	4096 ± 1536	4009 ± 1321	87	0.734
Light (min)	3134 ± 655	3148 ± 757	3119 ± 575	29	0.151
Moderate (min)	263 ± 115	292 ± 123	233 ± 98	59	0.0001 *
Vigorous (min)	67 ± 56	69 ± 53	64 ± 59	6	0.134
MVPA	48 ± 21	52 ± 23	43 ± 19	10	0.0001 *
Step Counts	8800 ± 2066	8996 ± 2322	8603 ± 1758	393	0.0453 *

*-statistical significance.

**Table 3 ijerph-17-01375-t003:** Moderate to Vigorous Physical Activity classification.

	MVPA Classifications n (%)		*p*	Odds Ratio
	≥60 min	<60 min		
Gender			0.0001 *	0.37
Girls	31 (8.4)	154 (41.5)		
Boys	65 (17.5)	121 (32.6)		
BMI classification			0.0045 *	NA
Underweight	4 (1.1)	26 (7.0)		
Healthy body mass	77 (20.8)	219 (59.0)		
Overweight	5 (1.3)	23 (6.2)		
Obese	10 (2.7)	7 (1.9)		
Step counts classification			0.0001 *	33.68
≥12,000	19 (5.1)	2 (0.5)		
<12,000	77 (20.8)	273 (73.6)		

*-statistical significance.

**Table 4 ijerph-17-01375-t004:** Generalized linear models for MVPA and steps counts.

	MVPA [minutes/day]			Step Counts		
	β	95% CI	*p*	β	95% CI	*p*
Gender	−0.22	(−0.32, −0.12)	0.0001 *	−0.11	(−0.21, 0.01)	0.051
Age	0.06	(−0.06, 0.17)	0.351	0.09	(−0.03, 0.21)	0.155
Body mass	−0.97	(−2.05, 0.10)	0.075	−0.62	(−1.73, 0.50)	0.278
Body height	0.64	(−0.01, 1.29)	0.053	0.32	(−0.35, 0.99)	0.348
BMI classification	0.75	(0.01, 1.46)	0.038 *	0.46	(−0.28, 1.19)	0.223

*-statistical significance.

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
