# Peer review of "Objectively Assessed Physical Activity of Preschool-Aged Children from Urban Areas"

_ijerph, 2020, doi:10.3390/ijerph17041375_

Round 1
Reviewer 1 Report
Review of the manuscript entitled "Objectively assessed physical activity of preschool age children from urban areas
Abstract:
The abstract is perfectly adapted to the basic characteristics of scientific manuscripts. Perhaps, one of the key words should be removed as young children and preschool can mean the same thing.
Introduction:
The paper presented by the authors has a sufficient scientific consistency since it makes an accurate review of the research topic. In addition, the authors coherently delimit the objectives of the research.
Material and methods:
The modifications made by the authors strengthen the manuscript.
A question arises with the calibration of the accelerometers, why have they used 5-second periods instead of 10 seconds?
Results:
Now they are expressed correctly and make reading and understanding easier. The effort made is appreciated.
Discussion and conclusions:
These are the most relevant sections of the manuscript. They are sections that present cohesion with the rest of the manuscript thanks to the improvements made.
Author Response
Thank you very much for the review of the original article entitled “Objectively assessed physical activity of preschool age children from urban areas”.
Referring to an earlier review commentary, we used in the keywords young children as children from kindergarten and preschool as an institution that meets important educational goals, among others. to meet a minimum of physical activity.
Based on reviewers' comments, we decided to perform new calculations. Data was collected in 5s epochs, which is better and more accurate for the spontaneous and intermittent activities of preschool children as used previously with a similar
sample [28. Vale, S.; Silva, P.; Santos, R.; Soares-Miranda, L.; & Mota, J. Compliance with physical activity guidelines in preschool children. J. Sports Sci. 2010, 28(6), 603–608. doi:10.1080/02640411003702694].
The authors also sent the article to a native speaker for correction in English.
Reviewer 2 Report
Given the substantial edits, I believe the paper reads much stronger. However, prior to publication, I would encourage the authors to give the paper a solid read-through to ensure any final typos and errors are caught. For example, I believe there are some words missing (or incomplete words?) in line 244.
Author Response
Thank you very much for the review of the original article entitled “Objectively assessed physical activity of preschool age children from urban areas”.
The authors also sent the article to a native speaker for correction in English.
Reviewer 3 Report
As mentioned in my first review, I always miss a clear statistic testable hypothesis yet. The other comments of myself are more or less included.
Author Response
Thank you very much for the review of the original article entitled “Objectively assessed physical activity of preschool age children from urban areas”.
We added the following research hypotesis:
"Based on previous studies [Janssen, I.; Leblanc, A.G. Systematic review of the health benefits of physical activity and fitness in school-aged children and youth. Int J Behav Nutr Phys Act. 2010, 7, 40] we hypothesizes that most of subjects not meet WHO physical activity recommendation. We also assume that gender and BMI will significantly differentiate physical activity level of preschool children".
The authors also sent the article to a native speaker for correction in English.
This manuscript is a resubmission of an earlier submission. The following is a list of the peer review reports and author responses from that submission.
Round 1
Reviewer 1 Report
Revision of the manuscript entitled "Objectively assessed physical activity of preschool 2 age children from urban areas".
In the keyword section, keywords such as physical activity and kindergarten are missing.
The introduction is rather brief. It would be good to update some more current quotes.
The methodology is adequate, although it would be possible to indicate the time zone in which the anthropometric measurements have been taken.
Results
It would have been interesting if they had made a comparison between the BMI classification and the MVPA classification.
The discussion is well founded although current research can be used to contrast the data.
The conclusions are appropriate to the manuscript.
Reviewer 2 Report
Thank you for the opportunity to review. This paper adds some important insights into the PA levels of children living in urban Poland. However, I think the overall structure of the manuscript (incomplete paragraphs), stronger objective statements and rationale, as well as discussion of findings of implications is required. Further, the methods require work - the accelerometry approaches utilized (i.e., epoch length, consecutive zeros, etc.), as well as the PA guidelines, require updating for the specific age group. The most "current" methodological approaches have not been applied.
The novelty of this work could be more strongly emphasized as well.
Should this paper undergo a major overhaul and address the above mentioned issues, I think it will make a much stronger contribution to the literature.
Respectfully submitted.
Reviewer 3 Report
Objectively assessed physical activity of preschool age children from urban areas
Physical activity of children decreased not only in all industrial countries over more than the last 10 years. Children with a lower level of physical activity have a higher risk either to get obese or have muscle and bone related health problems or both later in life. Developing intervention program minimizing the risk is urgent. So it is important to analyze the current status in a population. The authors make a first step here for Poland, indeed.
However, I have doubts that it is publishable as a scientific paper in the presented form.
In the introduction the authors describe the aim of the study, but a clear statistical testable hypotheses is not formulated. What want to find they out? Do the authors assume a difference between the sexes and/or between the age groups or not; if yes or not on which theoretical basic. That should be of every scientific study the start.
Some remarks:
- An explanation of MVPA – Index is completely missing. Which variables are involved? How is the calculation?
- The authors measured height and weight and calculated out of them the BMI. These are the minimal used anthropometric variables commonly.
- The authors should try to find out, if there exist an association between physical activity and BMI depending on age and sex using the statistical method of linear mixed models.
- Since at least 10 years ago a German working group used for same scientific questions an additional anthropometric variable the elbow breadth (as Frame Index = elbow breadth/height (Frisancho 1990)) and published the results. The Frame Index is easy to determine and describes the external skeletal robusticity. Additional with BMI we have an impression about the body composition (see in book Auxology ed. by Hermanussen 2013). Frame index is stronger in pre school children with a two year additional sports intervention program comparable to children without intervention program (Scheffler et al 2007), whereas the BMI is not different. In schoolchildren was shown that the daily physical activity as higher influence on body composition than attending e.g. sport clubs (Rietsch et al. 2013). International reference values of Frame-Index, which includes also data from Polish children were also published (Mumm et al. 2018).
- It is really pity that a study with the focus of physical activity, BMI and health started in 2018 not included the complex analyses of body composition.
- The authors should discuss why the differences between age and sex are there plausible or not. What are possible explanations. Are the higher value in the elderly group of MVPA (> 60 min) in both sexes?
- To that extent, the limit of the study is not a missing of a rural population. The limit is the developed methods and the bad statistical analyses.
Some minor remarks:
- Where are the kcal – specification from? What do they tell?
- The authors compared sex-groups and not gender groups. Gender is a social construct. Please do not switch between both term and use sex.
Reviewer 4 Report
The authors present an important paper that fills a gap in the literature. Specifically, little objective measurement of physical activity are available for young children and more particularly preschoolers.
Suggestions/Comments for the Abstract:
Overall, not enough results or discussion; introduction and methods can be condensed. Conclusion is not clearly supported by the data/purpose of the study.
Line 11: missing ‘measuring’ (i.e., the reliability of measuring daily PA)
Lines14-16: repetitive
Line 18: groups? What groups? age
Suggestions/Comments for the Introduction:
Although a relatively sound context was presented to support the research undertaken, it is lacking. Specifically, it is important to recognize the various contributors to children’s physical activity, i.e., active transportation (mentioned), free play, organized play/sport (e.g., swimming lessons, football, etc.), and physical education (assume in Kindergarten this is a mandatory component). It is also important to recognize the various individual responsible for providing children with opportunities to be physical active, i.e., parents, teachers, physical activity/sport program providers, etc.
Screen time could be recognized as a competitor for physical activity time – at home and at school.
You might include physical activity guidelines for under the age of 5 given your sample includes 4-5 year olds. Canada has established 24 hour movement guidelines (https://csepguidelines.ca/). It does not seem appropriate to have the same expectations for movement for 4 yr olds when the guidelines listed are not intended for them.
Line 49: 12,000 steps is considered the equivalent to 60 min MVPA and is not a high level of PA – please correct.
To get to your conclusions, it is important to make the connection between measuring PA, creating interventions to promote PA, and then measuring the effectiveness of them.
Suggestions/Comments for Materials and Methods:
Lines 67-68: what is meant by ‘comprehensive information given to the participants’? The participants were 4-7 year olds? How were they given the opportunity to ‘opt out’ as 15 did?
Lines 75-79: reorganize to list number of participants first. I.e., Of the 340 children with parental consent, 15 opted out of the study. Of the remaining 325 participants (children who provided accelerometer data) 61 were excluded b/c …
Line 82: What are standard conditions?
Line 83: weight was measured via a body composition analyzer?
Line 89: rationale for children to be fasting prior to these measurements?
Lines 91-93 are repetitive
Overall description of accelerometers and how they work could be improved. In particular connect to measuring PA in children, esp young children whose PA is often erratic. Does 15 sec epoch cover the quick switch from high to low intensity PA for them?
Line 101: participants were instructed to wear the accelerometer 24 hours per day? While sleeping too? Given that only a wear time of 8 hours and 20 min was needed, this requires clarification.
Lines 112-113: what is meant by ‘accompanied by a pedagogue’?
Line 113: is this referring to the ‘comprehensive information about the study’ from line 68?
How was PA data organized in relationship to meeting PA guidelines? Specifically, were daily minutes of MVPA averaged and that average used to indicate meet/do not meet the 60 min MVPA guideline? Vs. were daily values for PA used to determine the number of days meeting PA guidelines? The same questions can be used regarding 12000 steps? Clarity is needed here. Both methods have been used (see https://www.participaction.com/en-ca/resources/report-card), the reader needs to know which one.
Further, were individuals who only provided 4, 5, or 6 days of data used in determining the percentage of children meeting the PA guidelines? 12000 steps?
Suggestions/Comments for Results:
What is the importance of including the kcals data? How valid are these data for children 4-7 years – this is not included in the methods? Why would kcals be different between 4-5 and 5-6 years olds when there is no sig difference in their PA? or sedentary behaviours?
Tables 2 and 3 could be combined.
Rationale for comparing 4 & 5 yrs olds to 6 & 7 yr old vs. an ANOVA and comparing 4 groups?
Suggestions/Comments for Discussion and Conclusion:
Lines 163-166: what about light physical activity? Participants had about 4 hours of light PA?
Lines 195-199: why are ranges of steps recommended?
Line 199: accelerometers have been used to measure PA objectively for a long time – not sure they can be described as ‘more popular’ right now.
Line 204: large group?
Line 208: what do you mean by ‘no interventions’ in this study?
This discussion is missing some thoughtful comments regarding the discrepancy in the percentage of children meeting the 60 min MVPA vs. 12000 steps. Further, the discussion should include some comments regarding the nature of preschoolers PA and the implications on measurement.
Other limitations to include regarding the sample used – socioeconomic status? Race? And overall representativeness of the population?
To reach the conclusion made regarding physical activity promotion, context regarding baseline measurements from which to compare should be made.
Again, emphasis for promoting PA for preschoolers should be for the various individuals involved (parents, teachers, program providers, infrastructure regarding active transportation, playgrounds, etc.) such that opportunities for preschoolers’ physical activity can be enhanced at home, school, and in the community.